# Colistin Resistance in *Acinetobacter baumannii:* Molecular Mechanisms and Epidemiology

**DOI:** 10.3390/antibiotics12030516

**Published:** 2023-03-04

**Authors:** Katarina Novović, Branko Jovčić

**Affiliations:** 1Institute of Molecular Genetics and Genetic Engineering, University of Belgrade, Vojvode Stepe 444a, 11042 Belgrade, Serbia; 2Faculty of Biology, University of Belgrade, Studentski Trg 16, 11000 Belgrade, Serbia

**Keywords:** *Acinetobacter baumannii*, colistin resistance, *lpx*, *pmr*, *mcr*, LPS, lipid A, phosphoethanolamine transferase, epidemiology

## Abstract

*Acinetobacter baumannii* is recognized as a clinically significant pathogen causing a wide spectrum of nosocomial infections. Colistin was considered a last-resort antibiotic for the treatment of infections caused by multidrug-resistant *A. baumannii*. Since the reintroduction of colistin, a number of mechanisms of colistin resistance in *A. baumannii* have been reported, including complete loss of LPS by inactivation of the biosynthetic pathway, modifications of target LPS driven by the addition of phosphoethanolamine (PEtN) moieties to lipid A mediated by the chromosomal *pmrCAB* operon and *eptA* gene-encoded enzymes or plasmid-encoded *mcr* genes and efflux of colistin from the cell. In addition to resistance to colistin, widespread heteroresistance is another feature of *A. baumannii* that leads to colistin treatment failure. This review aims to present a critical assessment of relevant published (>50 experimental papers) up-to-date knowledge on the molecular mechanisms of colistin resistance in *A. baumannii* with a detailed review of implicated mutations and the global distribution of colistin-resistant strains.

## 1. Introduction

Colistin (polymyxin E) is a nonribosomally synthesized polycationic peptide that belongs to the class of polymyxin antibiotics, of which only two are used clinically: polymyxin B and colistin [1]. Colistin was introduced into clinical practice in the 1950s, but its use in human medicine was mainly limited to the treatment of pulmonary infections caused by multidrug-resistant (MDR) Gram-negative pathogens in patients with cystic fibrosis due to nephrotoxicity and neurotoxicity [2,3]. However, the widespread use of colistin in animal feed production has been maintained in developing countries and poses a major public health risk [4]. The rise of MDR, extensively drug-resistant (XDR), and pan drug-resistant (PDR) strains of Gram-negative bacteria has sparked interest in the revival of antibiotics, such as colistin, which can be used as a last resort [5,6,7].

Colistin is a mixture of the cyclic decapeptide colistin A and B with a fatty acid chain (6-methyl-octanoic acid in colistin A or 6-methyl-heptanoic acid in colistin B) linked by an alpha-amide bond. The amphiphilic surfaces of colistin, which allow detergent-like activity on bacterial membranes, are formed by the N-terminal fatty acyl chain, D-Leu-Leu (hydrophobic), and three cationic amino acids (hydrophilic) [8,9]. Two forms, colistin sulfate for oral administration and colistimethate sodium for parenteral formulations, are currently commercially available.

Colistin is positively charged, therefore, it interacts electrostatically with the negatively charged phosphate groups of lipid A, the lipopolysaccharide (LPS) component of Gram-negative bacilli outer membrane [10]. After the initial interaction, colistin displaces the divalent calcium and magnesium cations that affect the three-dimensional structure of LPS. In the next step, colistin inserts its hydrophobic terminal acyl fatty chain, leading to disruption and permeabilization of the outer membrane. When permeabilization occurs, colistin penetrates the outer membrane and alters the integrity of the phospholipid bilayer of the inner membrane, causing intracellular material to leak out and leading to cell death [11] (Figure 1A). Therefore, colistin is considered a bactericidal antibiotic.

This review aims to provide a comprehensive insight into the clinical significance of *A. baumannii,* the molecular mechanisms of colistin resistance, and the epidemiology of colistin-resistant strains, as well as an overview of recent advances in the field.

## 2. Clinical Significance of *Acinetobacter baumannii*

*A. baumannii* is recognized as a clinically significant pathogen causing a wide spectrum of nosocomial infections, especially in vulnerable patient groups [12]. These groups include intensive care unit (ICU) patients, patients with prolonged hospitalization in long-term care facilities, patients undergoing surgeries, central vascular catheterization, tracheostomy, and enteral hemorrhage, and low birth weight neonates [13,14,15,16,17]. The literature data on nosocomial *A. baumannii* infections are mainly based on reports of outbreaks [18]. These outbreaks are usually due to contamination from common sources or cross-infection, and frequent serial or overlapping outbreaks could be observed once *A. baumannii* was introduced into a clinical setting with a single strain dominating each outbreak [19,20,21]. Community-acquired *A. baumannii* infections are less common (mainly pneumonia and bacteremia) and have a more severe course than nosocomial infections and are generally considered fulminant. These infections occur mainly in elderly male patients in association with alcoholism, diabetes renal disease, and chronic obstructive pulmonary disease [22,23,24]. Mortality rates associated with *A. baumannii* infections vary considerably depending on concomitant diseases, length of hospital stay, demographic characteristics, and antibiotic susceptibility of the strains causing the infection, generally ranging from 12 to 50% [19,25,26,27,28]. Of particular importance is the difficulty in distinguishing *A. baumannii* colonization from infection. The recognized risk factors for infection were age, total number of hospitalized wards, absolute neutrophil count, and C-reactive protein (CRP) [29].

Bacteria of the genus *Acinetobacter* are considered ubiquitous microorganisms, obtained from various environments, including soil, rivers, and wastewaters. Although *A. baumannii* reservoirs have been reported in the environment outside hospitals [30], the natural habitats of clinically relevant strains remain unclear [31].

*A. baumannii* possesses extraordinary plasticity that allows it to adapt to a variety of living conditions, enabling its success as a nosocomial pathogen [32]. The ability of *A. baumannii* to adapt to the challenges of the hospital environment is considered to be the major factor in its pathogenicity. In addition, the strain-dependent differential regulation of virulence genes, the large number of transcriptional regulators compared to other *Acinetobacter* species, and the synergy of multiple genes encoding virulence factors are thought to contribute to the virulence potential of *A. baumannii* [32,33,34].

A burning issue in the biology of *A. baumannii* is the global spread of MDR strains. The increase in MDR strains is driven by both intrinsic and acquired antibiotic resistance mechanisms. Strains of *A. baumannii* are capable of upregulating intrinsic mechanisms of antibiotic resistance, which, in conjunction with the acquisition of new resistance genes through horizontal gene transfer, contributes to the spread and diversity of the *A. baumannii* resistome. A variety of intrinsic resistance mechanisms of *A. baumannii*, such as beta-lactamases, multiple drug efflux pumps, changes in membrane-associated proteins, ribosomal methylation, and enzymes that recognize multiple antimicrobials as substrates, have been described previously [35].

In addition to intrinsic resistance, gene flow and horizontal transfer have been shown to be another important driver of antibiotic resistance genes in *A. baumannii* [36]. These processes resulted in observable, significant variation in the resistome within different lineages, and antibiotic resistance was shaped by phylogeny, resulting in what has been termed an open resistome [36].

*A. baumannii* is considered intrinsically resistant to penicillins and cephalosporins [37]. The resistance of *A. baumannii* to beta-lactams is significant because penicillins, cephalosporins, carbapenems, and monobactams are the first-line therapeutics for the treatment of infections caused by *A. baumannii*. Inherent in all *A. baumannii* are chromosomally encoded cephalosporinases (formerly *bla*_AmpC,_ now referred to Acinetobacter-derived cephalosporinase, ADC). Insertion of IS*Aba1* or IS*Aba125* sequences upstream of genes encoding ADC cephalosporinase induces its overexpression by providing stronger promoters [38,39,40]. The ADC enzymes may confer extended-spectrum resistance to beta-lactams [41,42,43]. In addition, covalent modification (dephosphorylation) of ADC enzymes could alter their substrate specificity and lead to imipenem resistance [44]. Several other beta-lactamases, such as extended-spectrum beta-lactamases (ESBLs) (including TEM, SHV, CTX-M, PER, VEB, and GES), metallo-beta-lactamases (MBLs) (including IMP, VIM, GIM, and NDM), and oxacillinases (OXAs) (including OXA-23-like, OXA-24-like, OXA-51-like, and OXA-58-like) are commonly found in *A. baumanni* clinical isolates [45,46]. Resistance to beta-lactams could result from changes in the permeability of the cell to the antibiotic, usually due to changes in outer membrane proteins such as CarO, OmpA, and Omp33-36 porins [47,48,49,50]. It has been found that overexpression of the AdeABC efflux pump synergistically with the aforementioned beta-lactamases in *A. baumannii* leads to carbapenem and cephalosporin resistance [51,52].

Colistin is considered one of the last therapeutic options for the treatment of MDR *A. baumannii* infections and is used as rescue therapy for severe infections. Colistin resistance poses a greater risk of excess patient mortality [53,54]. The published data show that the prevalence of colistin resistance is higher in Southeast Asia and Eastern Mediterranean countries than in other regions of the world, with an overall value of 11.2% (Germany 0.2%, United Kingdom 2.3%, India 8.2%, China 11.8%, and Lebanon 17.5%) [55].

## 3. Molecular Mechanisms of Colistin Resistance in *A. baumannii*

Colistin, as a positively charged peptide, exerts its antibacterial effect via electrostatic interactions with negatively charged lipid A, a component of LPS [56]. Accordingly, two main mechanisms of colistin resistance have been described in *A. baumannii:* the complete loss or modifications of the target LPS, leading to abolishing or reducing its negative charge [57]. The complete loss of LPS results from inactivation of the first three genes of the lipid A biosynthetic pathway (*lpxA*, *lpxC,* and *lpxD* genes) [58], whereas the modification of LPS occurs through the addition of phosphoethanolamine (PEtN) moieties to lipid A by the *pmrCAB* operon-encoded enzymes [59]. Although 4-amino-4-deoxy-L-arabinose (L-Ara4N) modification of LPS has been described as a more common and effective colistin resistance mechanism compared to PEtN LPS modification in diverse Gram-negative pathogens (*Salmonella enterica*, *Klebsiella pneumoniae, Escherichia coli*, and *Pseudomonas aeruginosa*), it was absent in *A. baumannii* [57]. In addition to chromosome-mediated mechanisms, plasmid-mediated colistin resistance encoded by *mcr* genes has been recognized as a major driver of rapid dissemination by horizontal gene transfer among pathogenic Gram-negative bacteria, including *A. baumannii* [60] (Figure 1B).

### 3.1. Loss of LPS Structure

The first observation that LPS deficiency causes colistin resistance in *A. baumannii* was made by Moffat and coauthors [58]. Laboratory-induced colistin-resistant *A. baumannii* derivatives contained mutations in one of the first three lipid A biosynthetic genes (*lpxA*, *lpxC,* or *lpxD*) (Figure 1B). Although, these mutations ranged from single nucleotide mutations to large deletions (up to 445 nucleotides), they all resulted in complete loss of LPS. Moreover, disruption of the *lpxD* gene by insertion of an IS element similar to the IS*X03* element (IS*4* family) was described in a colistin-resistant clinical isolate [58]. Shortly thereafter, the same team found that IS*Aba11* inactivated the *lpxC* and *lpxA* genes in colistin-resistant derivatives of *A. baumannii* ATCC19606 [61]. In subsequent studies, the insertion of IS*Aba1* or IS*Aba11* within the *lpxC* gene was described as a common event in colistin-resistant *A. baumannii*. As the disruption of the *lpxC* gene occurred in the same region (321–420 nt) in different isolates, it was suggested that this region might represent a hot spot for the integration of ISs [61,62,63,64,65,66,67,68]. Colistin resistance in *A. baumannii* has also been associated with various nucleotide substitutions, deletions, and insertions in all three lipid A biosynthetic genes (*lpxA*, *lpxC,* or *lpxD*) that cause frameshifts or result in truncated proteins that impair lipid A biosynthesis. While the described mutational events in the *lpxA* gene are not site-specific, non-synonymous mutations in the *lpxC* (P30L or S, N287D) and *lpxD* (E117K) genes were previously found to be present in colistin-resistant isolates from different origins [58,69,70,71,72,73,74]. Although the amino acid substitutions N287D (*lpxC*) and E117K (*lpxD*) were detected in both colistin-susceptible and colistin-resistant isolates, it is possible that these alterations in combination with a mutation in the *pmrCAB* operon have a synergistic effect leading to colistin resistance [69,70,71,72,73,74]. In addition, the downregulation of *lpxACD* expression has been observed in some colistin-resistant *A. baumannii* isolates, leading to decreased LPS production [68,73,74,75,76].

At the time when LPS deficiency was described as the mechanism causing colistin resistance in *A. baumannii,* this discovery was surprising because LPS biosynthesis was thought to be essential for the viability of Gram-negative bacteria [77]. So far, survival without LPS has been described only in a few species, such as *Neisseria meningitidis*, *Moraxella catarrhalis*, and two *Acinetobacter* species (*A. baumannii* and *A. nosocomialis*) [78,79,80]. Although this mechanism ensures a high level of colistin resistance [58,61,65], the frequency of mutations in the *lpxACD* is lower compared to changes in the *pmrCAB* operon in colistin-resistant *A. baumannii* clinical isolates [66,81,82]. The proposed explanation for the lower prevalence of LPS-deficient colistin-resistant mutants in clinical settings could be the significant negative impact of LPS loss on fitness and virulence, as well as the susceptibility of these isolates to various antibiotics and disinfectants. This was supported by the findings that the *lpx* mutants grew more slowly compared to the parental wild-type strains in vitro [64,66,68,81,83], while in vitro and in vivo competition tests showed significant fitness costs of colistin resistance [81]. Determination of the pathogenicity of the *lpx* mutants also revealed lower cytotoxicity (A549 human alveolar epithelial cells) and attenuated virulence of these strains in the animal models (*Caenorhabditis elegans, Galleria mellonella,* and mouse) compared to wild-type or even *pmrB* mutants [63,66,81]. As expected, the absence of LPS on the cell surface resulted in weak stimulation of neutrophils and, consequently, lower production of reactive oxygen species (ROS) and pro-inflammatory cytokines [66,83]. Nevertheless, the *lpx* mutants were more prone to killing mediated by neutrophils compared to the wild type since they were more susceptible to neutrophil-secreted lysozyme [83]. Moreover, reduced virulence of LPS-deficient *A. baumannii* in the host could also be explained by reduced biofilm formation, surface motility, as well as poor growth under iron limitation [66,83]. Finally, another disadvantage of LPS loss for the bacterial cell is evident in the increased susceptibility to various clinically used antibiotics, especially antibiotics used in the therapy of *A. baumannii* infections (ceftazidime, imipenem, meropenem, tigecycline, ciprofloxacin, amikacin, and rifampin), and various disinfectants (phenol-based disinfectants, quaternary ammonium disinfectants, sodium dodecyl sulfate, benzalkonium, chlorhexidine, deoxycholate, and ethanol) [58,63,64,65,66,68,83].

### 3.2. PEtN Modification of LPS Structure

#### 3.2.1. PmrCAB and EptA

The modification of LPS is a commonly described mechanism for acquired colistin resistance in Gram-negative bacilli. In *A. baumannii*, PetN is added to the 4′-phosphate or 1-phosphate group of lipid A, reducing the negative charge of this LPS component and the binding affinity of colistin [57] (Figure 1B). This type of colistin resistance is predominantly mediated by mutations in genes encoding the PmrAB two-component system, resulting in the overexpression of the phosphoethanolamine transferase PmrC [84] (Figure 1B). The most common and diverse amino acid changes associated with colistin resistance in *A. baumannii* were detected in the PmrB protein (Table 1). Since Adams et al. [59] observed that mutations in the *pmrB* gene can cause high colistin resistance (MIC greater than 128 µg/mL) in laboratory-induced *A. baumannii* derivatives, numerous studies have described the presence of altered PmrB proteins in colistin-resistant clinical isolates or in vitro-derived derivatives of *A. baumannii* (Table 1). Although these nonsynonymous mutations were detected in all domains of PmrB, the greatest number were located in the histidine kinase A (HisKA) domain (predominantly at positions 226, 227, 232, 233, 235, and 263) (Table 1), which is responsible for autophosphorylation and the transfer of the phosphoryl group to the PmrA response regulator [85]. Accordingly, *pmrB* mutations could lead to the constitutive activation of PmrA, resulting in increased expression of the *pmrCAB* and resistance to colistin [59]. In addition, previous studies reported frequent amino acid substitutions of PmrB at position 170 (P to Y, L, Q, or S) (Table 1) and 315 (G to D, S, or V) in colistin-resistant isolates [68,70,76,84,86]. Although Oikonomou and coauthors [69] described the PmrB mutations (A138T, A226V, and A444V) repeated in colistin-resistant *A. baumannii* [70,72,73,74,76,84,85,86,87,88,89,90,91] as not responsible for colistin resistance, the involvement of A138T and A226V in this phenomenon should not be excluded. Indeed, the amino acid change at position 226 (A to V) in PmrB alone or in combination with A138T enabled high colistin resistance (64 or 128 and 256, respectively) in the tested isolates [84,88]. The amino acid substitutions within the receiver domain (REC) of the PmrA response regulator have also been described in *A. baumannii* as resistant to colistin (E8D, D10N, M12I, K, or V, I13M, or S, A14V, I18T, L20F, G54E, A80V, D82G, P102H, or R, F105L) [59,68,69,72,84,86,89,90,92,93,94,95,96,97,98]. Some of the PmrA mutations alone (G54E) or in combination with mutations in other genes (P102R) can confer significantly high colistin resistance to *A. baumannii* (>256 µg/mL or 512 µg/mL) [97,98]. To date, little data are available on the relationship between PmrC amino acid changes and colistin resistance. A comparison of PmrC amino acid sequences between colistin-susceptible and colistin-resistant isolates revealed rare changes and mostly at different positions [65,69,72,73,74,75,76,84,89,95,97]. In the study conducted by Nurtop and coauthors [72], the two most commonly described mutations in the *pmrC* gene (resulting in I42V and L150F) were found to be associated with an increased expression of the *pmrA* and *pmrC* genes and, consequently, colistin resistance. The PmrC substitution R109H, detected in colistin-resistant *A. baumannii* isolates in two previous studies [69,72], was associated with colistin resistance along with a mutation in the *pmrA* gene [69]. In addition, it was observed that the PmrC alteration R125P in combination with changes within the PmrB protein had a synergistic effect on colistin resistance in *A. baumannii* [97]. In summary, mutations in the *pmrCAB* locus are recognized as gain-of-function mutations because they lead to PmrC overexpression and PEtN modification of lipid A, which, in turn, results in colistin resistance [84,99]. In addition to increased expression of PmrC as a mechanism of colistin resistance in *A. baumannii* [65,72,73,75,76,84,85,88,92,97,98,100], the upregulation of the *pmrA* and *pmrB* genes was found in some colistin-resistant isolates [59,71,96,101,102], but to a much lesser extent [68,72,73,75,76,84,92,98]. Although this observation is to be expected as these genes are part of the same operon as the *pmrC* gene (*pmrCAB*), there are cases where no correlation was found between PmrAB and PmrC overexpression [72,73,76]. In addition, Lesho and coauthors [92] noted the overexpression of another *pmrC* homolog (*eptA*, ethanolamine phosphotransferase A) in some colistin-resistant *A. baumannii* isolates. Detailed analysis revealed that the gene encoding for EptA was detected only in isolates belonging to the international clone 2 (IC2), was found in ≥3 copies in a single isolate, and was distant from the *pmrCAB* operon in *A. baumannii* genomes [88,90,92]. Although the presence of the *eptA* gene in the bacterial genome alone does not confer resistance to colistin, the integration of IS*Aba1* upstream of the *eptA* gene could lead to increased expression of this enzyme [88] (Figure 1B). In contrast, Gerson et al. [100] found the presence of IS*Aba1* upstream of the *eptA* gene in colistin-susceptible and colistin-resistant counterparts, but only in isolates with mutations in the *eptA* gene (R127L) and IS*Aba1* (A→T in position 1091) was overexpression of EptA detected. Interestingly, a previous study showed that disruption of the gene encoding the global regulator H-NS by IS*Aba125* causes high colistin resistance in *A. baumannii* through increased expression of the *eptA* gene in this mutant strain [103].

A negative correlation was found between PmrAB-related colistin resistance and the fitness and virulence of *A. baumannii* in the host. The colistin-resistant *A. baumannii* isolates showed lower fitness in vitro and in vivo and reduced virulence potential in animal models of infection compared to their colistin-susceptible parental strains [62,92,93,104,105,106,107,108]. This could be explained by the downregulation of metabolic and antioxidant proteins, virulent porins OmpA and CarO, and the system responsible for biofilm formation in colistin-resistant *A. baumannii* [107,109,110]. In addition, some studies reported a correlation between colistin resistance and decreased biofilm formation ability [108,110]. In contrast to the initially reported negative correlation, additional studies showed unchanged fitness [63,64,68,100,111,112] and pathogenicity of colistin-resistant *A. baumannii* [63,81,100,111]. Interestingly, two studies described the emergence of colistin resistance in *A. baumannii* isolated from patients exposed to colistin therapy and the subsequent disappearance of this resistance after the discontinuation of colistin [111,113]. Durante-Mangoni and coauthors [111] observed that colistin-resistant *pmrB*-mutated isolates were comparable to wild type in in vitro and in vivo assays, whereas Snitkin et al. [113] hypothesized that resistant isolates were outcompeted by colistin-susceptible isolates due to lower in vivo fitness costs. In addition, a comparison of five longitudinal colistin-resistant *A. baumannii* isolates from the same patient indicated an increase in growth rate as well as virulence in the mouse lung infection model during colistin therapy [114]. Jones and coauthors [114] explained this phenomenon by more pronounced resistance to ROS in late colistin-resistant isolates. Overall, these data suggest that no clear conclusion can be made about the correlation of colistin resistance due to *pmrAB* mutations and biological costs in *A. baumannii*. Although some *pmrAB* mutations responsible for colistin resistance initially appeared to be maladaptive to bacterial cells, prolonged exposure to the selective agent (colistin) may have allowed the emergence of compensatory changes at different regulatory levels and remedied a deficit in fitness and virulence [63,104,113,114]. In addition, in this type of research, the genetic background should be taken into account as the results obtained from different isolates containing the same PmrB alteration P233S were different [107,108,111,112]. The studies comparing the behavior of the *pmrAB* mutants with *lpxACD* mutants have undoubtedly confirmed that the LPS modification causes lower fitness and virulence costs than LPS deficiency [63,64,81]. Most studies that examined colistin-resistant *A. baumannii* showed that PmrAB alterations had no significant impact on the antibiotic resistance profile of these isolates [64,68,69,84,92,112]. Consistent with the above observations, a systematic review concluded that LPS modification mediated by the *pmrAB* mutations is the major in vivo mechanism of colistin resistance [82].

#### 3.2.2. Plasmid-Mediated Colistin Resistance

Since the first report of the phosphoethanolamine transferase-encoding *mcr* gene (*mcr-1*) in *E. coli* in China [119], the presence of this gene and its variants has been demonstrated in many Gram-negative bacteria distributed worldwide [60]. To date, ten different *mcr* gene families (*mcr-1* to *mcr-10*) with more than 100 variants have been described in bacteria isolated from animals, food, humans, and the environment [60,120]. In *A. baumannii,* the *mcr-1* and *mcr-4.3* variants are most commonly detected (Figure 1B). The *mcr-1* has been reported in clinical isolates from Asia (Pakistan, Iraq, and China) and Africa (Egypt) at sporadic frequency (*n* = 1–3) with the exception of samples collected from hospitals in Iraq (up to 89) [121,122,123,124,125,126]. The earliest *mcr-4.3*-positive isolate of *A. baumannii* was recovered from the cerebrospinal fluid of a patient with meningitis in Brazil in 2008 [127], which preceded the *mcr* discovery by Lui and coauthors [119]. Subsequently, *mcr-4.3* was detected in pig feces from a slaughterhouse in China [128] and in isolates from the Czech Republic [129,130]. The studies from the Czech Republic suggest that food imports from Latin America (frozen turkey livers from Brazil) and Asia (frog legs from Vietnam) may represent the primary route of transmission of *mcr*-carrying *A. baumannii* to Europe and thus to European hospitals [129,130]. As some studies showed that the recombinant expression of *mcr-4.3* in *E. coli* did not alter colistin MIC [131,132], while another study indicated that the heterologous expression of *mcr-4.3* could ensure colistin resistance through LPS modification in *A. baumannii* [127], it is not possible to draw a firm conclusion about its role in colistin resistance. Moreover, a comparative analysis revealed that the *mcr-4.3*-harbouring plasmids in *A. baumannii* share a common origin for this structure. It was found that these plasmids are untypable and cannot be transferred to other bacteria by conjugation or transformation [128,129,130]. Although *mcr-1* and *mcr-4.3* are predominant, other *mcr* variants have also been described in clinical and environmental samples of *A. baumannii*, as in a study from Iraq where the *mcr-2* and *mcr-3* genes were found. A large number of these isolates carry a single *mcr* gene or a combination of different *mcr* families (*mcr-1*, *mcr-2*, and *mcr-3*) [122]. Finally, it is important to highlight that most of the *mcr*-carrying *A. baumannii* isolates are MDR [121,122,124,125,126,127], and there are few antibiotic-susceptible isolates [128,129].

### 3.3. Other Mechanisms of Colistin Resistance

In addition to the aforementioned mechanisms of colistin resistance in *A. baumannii,* expulsion of the antibiotic by efflux pump systems is another mechanism of importance (Figure 1B). Lin and coauthors [133] demonstrated the contribution of the EmrAB efflux system to colistin resistance in *A. baumannii* using the Δ*emrB* mutant (Figure 1B). In addition, the upregulation of genes encoding protein components of efflux pumps (*adeI*, *adeC*, *emrB*, *mexB,* and *macAB*) was shown in colistin-resistant strains [67]. In addition, an amino acid substitution (N104M) in the gene encoding the toluene tolerance efflux pump (*ttg2C*) was found to be associated with high-level colistin resistance [70]. Further evidence for the role of efflux pumps in colistin resistance is the suppression of resistance in the presence of the efflux pump inhibitor (EPI), cyanide-3-chlorophenylhydrazone (CCCP) [134].

Another mechanism of colistin resistance in *A. baumannii* is associated with certain non-Lpx (lipo) proteins involved in the composition and maintenance of the outer membrane (OM) (*lpsB*, *lptD, vacJ*, *pldA*, and *A1S_0807*) [99,135]. The study conducted by Hood et al. [136] indicated that the loss of LpsB, a glycosyltransferase responsible for LPS core synthesis, leads to increased susceptibility to colistin and cationic host defense peptides, highlighting the role of this protein in *A. baumannii* virulence. Along with changes in the *pmr* and *lpx* genes, single mutations in the *lpsB* gene (H181Y and *241K) of colistin-resistant *A. baumannii* have been reported [108,137]. In addition, the final translocation of LPS from the cytosol to OM could be disturbed by mutations in the *lptD* gene, which has resulted in moderate polymyxin B resistance [138]. Colistin resistance of certain *A. baumannii* isolates analyzed by Nhu and coauthors [70] was attributed, in whole or in part, to amino acid substitutions of the OM lipoprotein VacJ (R166N and Q249T) and the phospholipase PldA (T200T). As VacJ, part of the ABC transporter system, and PldA are recognized as factors responsible for maintaining lipid asymmetry in OM, the proposed mechanism of this type of colistin resistance is OM disorganization due to *vacJ* and *pldA* mutations [70]. In addition, it has been observed that impaired lipid metabolism caused by a reduction in biotin synthesis could provide protection to *A. baumannii* during colistin exposure [136]. Recent studies using modern technologies (whole genome sequencing, RNA sequencing, and transposon-directed insertion site sequencing) have identified numerous genes (*Ab09_2943*, *ACICU_02910*, *ACICU_RS15345*, *A1S_1983*, *A1S_2024*/*ACICU_01043*, *A1S_2027*, *A1S_2230/ACICU_02436*, *A1S_2443*, *A1S_2928*, *A1S_3026*, *aroP_3*, *baeR*, *benP*, *betI_2*, *cho1*, *cobS*, *cobV*, *cysH*, *dcm*, *dnmT1*, *dtyMK*, *eno*, *filD*, *garK*, *glxK*, *hepA*, *iclR*, *lpsO*, *mdh*, *miaA*, *mlaC*, *mlaD*, *mlaF*, *mutY*, *mpsT*, *pgaB*, *pheS*, *pssA*, *pstS*, *ptk*, *rsfS*, *shlB_1*, *sseA*, *tmk*, *tst udg*, *ureC*, and *zndP*) whose sequence or expression in colistin-resistant *A. baumannii* was altered compared to colistin-susceptible strains [64,67,70,75,76,97,98,139]. The degree of association of these candidate genes with colistin resistance in *A. baumannii* should be confirmed experimentally in future studies.

### 3.4. Colistin Heteroresistance and Dependence

Antibiotic heteroresistance is defined as the presence of a resistant subpopulation within a population susceptible to a given antibiotic [140]. Since the first report of colistin heteroresistance in clinical isolates of *A. baumannii* from Australia [141], this phenomenon has been described in many studies with prevalence ranging from 1.84 to 100% [142,143,144]. Hawley and coauthors [142] found a higher rate of heteroresistance in isolates from patients treated with colistin, suggesting that previous colistin therapy may be a risk factor for the induction of heteroresistance. Data indicating resistance stability within the surviving subpopulation after cultivation under non-selective conditions were conflicting in different studies, suggesting a possible species-specific dependence [140,141,142,145]. Interestingly, Hong et al. [140] observed isolates that exhibited a heteroresistance phenotype only at low antibiotic concentrations alongside the typical colistin-heteroresistant isolates that emerged at exposure to high colistin concentrations. The previously described mechanisms of colistin heteroresistance in *A. baumannii* are the same as those of colistin resistance (LpxACD, PmrCAB, and efflux pumps) [73,140,143,145,146]. Amino acid changes in LpxC (S186R) and LpxD (N148K and T289I) were associated with partial loss of LPS in heteroresistant strains [143], while another study showed upregulation of the *pmrCAB* operon in combination with mutations in the *pmrA* and *pmrB* genes in resistant subpopulations of *A. baumannii* [146]. The overexpression of efflux pumps and the synergistic effect of EPIs and colistin against the resistant subpopulation of heteroresistant *A. baumannii* clearly demonstrated the involvement of efflux pumps in this phenotype [143,145]. Of particular concern is the fact that conventional susceptibility testing identifies heteroresistant isolates as susceptible to colistin, resulting in colistin treatment failure [143]. As population analysis profiling (PAP) is recognized as the gold standard for detecting heteroresistance, the introduction of the mini-PAP method with colistin at >2 mg/L into clinical practice has been recommended [147]. Moreover, the prevalence of heteroresistant isolates clearly exceeds the occurrence of colistin-resistant *A. baumannii* [148]. Moreover, under selection pressure, a resistant subpopulation of heteroresistant populations could become predominant and lead to a resistant cell population [145]. Accordingly, isolates identified as colistin-heteroresistant have been proposed for colistin-based combination therapy instead of colistin monotherapy [144]. Although the phenomenon of colistin heteroresistance has been studied mainly in *A. baumannii* of nosocomial origin, it has also been detected in samples from hospital wastewaters [73,149].

Another phenomenon observed in some colistin-susceptible *A. baumannii* isolates exposed to colistin is colistin dependence. After exposure to colistin, a colistin-dependent subpopulation of cells becomes dependent on this antibiotic for optimal growth [150]. Previous findings have suggested the colistin-dependent phenotype as a survival response to colistin pressure and an intermediate stage between colistin susceptibility or heteroresistance and even extreme resistance to colistin [65,150].

## 4. Epidemiology of Colistin-Resistant *A. baumannii*

Data providing information on the epidemiology of colistin-resistant *A. baumannii* are generally shown by MLST categorization (Oxford and Pasteur) of these isolates [151,152]. According to the less discriminating Pasteur scheme, colistin-resistant *A. baumannii* sequence type (ST) 2 isolates are found to be most prevalent ST associated with colistin resistance in *A. baumannii* and occur in all continents for which data are available (Europe, Asia, Africa, and North and South America) [73,74,75,76,86,88,89,90,91,97,107,153,154,155] (Figure 2). In addition, ST1 was detected in Europe and Africa [90,95,97,154,156], whereas other Pasteur STs occurred exclusively in specific continents (Europe—ST195, ST345, ST490, ST492, ST537, ST632, ST636, ST745, ST1421, and ST1816; Asia—ST1303; Africa—ST158 and ST570; North America—ST46 and ST94; South America—ST15, ST25, ST79, and ST730) [74,76,86,88,89,92,97,116,128,129,130,157] (Figure 2). In addition, ST1 has been transmitted both nosocomially [90,97], and by animals in Europe [95].

MLST typing according to the Oxford scheme revealed that some STs of *A. baumannii* resistant to colistin are distributed across different continents: ST92 (Asia and North America) [101,158], ST195 (Europe and Africa) [76,154], ST208 [76,86,90,97] and ST281 [75,86,97] (Europe and North America), and ST451 (Europe, Asia, and North America) [86,90,97,139,158] (Figure 2). Interestingly, ST208 has been suggested to be identical to ST92, with the typing result depending on the sequencing performed (high-throughput or Sanger, respectively) [86]. This suggests a significant dissemination of colistin-resistant *A. baumannii* in Europe, Asia, and North America. In Europe, the following STs have been reported in more than one study: ST208 [76,90,97]; ST281 [75,97]; ST425 [76,90,97]; and ST436, ST451, and ST1567 [90,97]. However, in North America and South America, only a single ST, ST451 [86,158] and ST233 [96,157] were detected, respectively. Additionally, some STs were detected only in single studies (ST113, ST141, ST191, ST218, ST227, ST231, ST233, ST236, ST282, ST369, ST375, ST387, ST502, ST747, ST944, ST1114, ST1557, ST1566, ST1633, ST1752, ST1809, ST1812, ST1837, ST1929, ST1962, and ST2571) [76,86,96,97,102,117,127,128,129,139,154,157,158,159,160] (Figure 2).

The literature search revealed a lack of data on the epidemiology of colistin-resistant *A. baumannii* STs in Australia and Oceania, pointing out the need for additional primary research to fill the existing knowledge gap.

## 5. Conclusions

*A. baumannii* has become a significant nosocomial pathogen because of its adaptability to healthcare settings, virulence characteristics, and ability to acquire antibiotic resistance. The increasing prevalence of MDR strains enhanced the use of colistin as rescue therapy, leading to the rise in colistin-resistance strains worldwide. The diversity of the colistin resistome in *A. baumannii* encompassing multiple mechanisms, including dissemination through horizontal gene transfer, requires thorough investigations that will provide comprehensive knowledge of this emerging pathogen and provide insights into the mechanisms of antibiotic resistance that will direct novel areas of research. Given the increasing prevalence of colistin-resistant strains, a reassessment of current therapeutic approaches, including alternatives to traditional antibiotics therapies, is strongly recommended. Promising results have been shown in vitro for cefiderocol (a molecule with an innovative mode of action), intravenous fosfomycin (in combination with cefiderocol), and combination therapy with sulbactam–durlobactam [161,162,163].

## Figures and Tables

**Figure 1 antibiotics-12-00516-f001:**
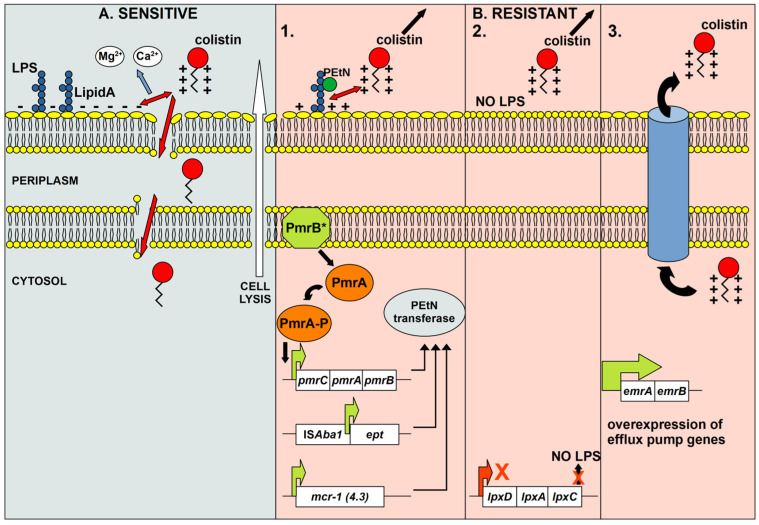
Schematic representation of the mode of action of colistin in susceptible cells (**A**) and the molecular mechanisms of resistance to colistin in resistant cells (**B**). A—Colistin causes lysis of susceptible cells due to induced disruption of the outer and inner membranes. Initial interactions between the positively charged moiety of colistin and the negatively charged phosphate groups of lipid A of LPS lead to the displacement of calcium and magnesium cations affecting the LPS structure. In the next step, the hydrophobic acyl fatty chain of colistin penetrates the outer membrane leading to its permeabilization. As a result of permeabilization, colistin penetrates the inner membrane and alters it integrity, leading to leakage of intracellular material and cell death. B—Resistance to colistin arises through several common mechanisms. 1. As a result of PEtN moiety addition to lipid A (mutations and overexpression of *pmrCAB*, *eptA*, or presence of plasmid-mediated *mcr* genes), the overall charge of the outer membrane changes so that colistin can no longer interact with lipid A of LPS. 2. Inactivation of the LPS biosynthetic pathway results in the absence of LPS, the target molecule for colistin. 3. Overexpression of specific efflux pumps leads to efficient extrusion of colistin, resulting in resistance.

**Figure 2 antibiotics-12-00516-f002:**
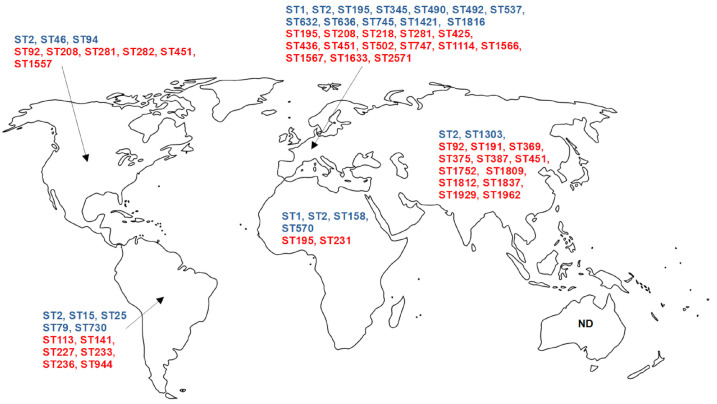
Global distribution of colistin-resistant *A. baumannii* STs according to published studies. Data for North America include the United States of America and Canada, while data for South America include data for the rest of the American continent. Pasteur MLST scheme—blue letters, Oxford MLST scheme—red letters. ND—no data available.

**Table 1 antibiotics-12-00516-t001:** Overview of PmrB sensor kinase mutations in colistin-resistant *A. baumannii*.

Domains	Amino Acid Mutation	Reference
TM1	L9_G12del	[100]
	T13N, T13A	[59,113]
	S14L	[115,116]
	S17R, S17G	[63,76,92,100]
	S17_F26dup	[63]
	I19F, I19del	[76,112]
	G21V	[96]
	F26fs	[117]
	A28V, A28T	[97,100]
PD	A32_E35del	[84]
	T42P	[97]
	Q43L	[88]
	K45Q	[97]
	H54Y	[97]
	D64V	[84]
	F65L	[97]
	K67R	[97]
	T68N	[74]
	I76L	[97]
	A80V	[84]
	H86N	[97]
	L87F	[115]
	H89L	[72]
	L93F	[97]
	L94W	[67]
	E99Q	[97]
	F103L	[76]
	Q110E	[97]
	I112V	[97]
	Y116H	[92]
	P119L	[86]
	I121F *	[101]
	Q129L	[87]
	R134C, R134S	[70,86,104]
	A138T	[69,72,73,74,85,87,88,89,91,116]
	E140V	[90]
	A142V	[85]
TM2	M145I, M145K	[113,115]
	L153F	[97]
	L160F, L160del	[76,84]
	I163F, I163N	[72,76]
	I164L, I164F	[72,76]
HAMP domain	R165S	[97]
	P170Y, P170L, P170Q, P170S	[65,67,84,96,97,107]
	L178F	[90]
	K179M	[97]
	S183F, A183T *	[76,101,116]
	E184K, A184V *	[65,101]
	E185K	[76]
	P190S *	[101]
	T192I *	[101]
	Y194S	[70]
	P200L	[91]
	L208F, L208R	[75,84,86]
	F209fs	[59]
	E210D	[97]
	R211S	[97]
HisKA	A224V	[72]
	A226V, A226T	[69,76,84,86,87,88,90]
	A227V	[59,65,68,85,96,104,115,117]
	Q228P *	[101]
	E229D	[70,72,91]
	R231L, R231T, R231I	[68,84,95]
	T232I, T232A	[68,76,86,92,100]
	P233T, P233S	[59,62,68,72,84,85,86,88,98,107,111,112,113,115,117]
	T235I, T235N	[63,68,84,85,86,98]
	L239S	[118]
	N256I	[84]
	A262P	[59]
	R263H, R263C, R263G,	[62,68,72,73,74,75,76,84,86,92,113,118]
	R263S, R263L, R263P	
	Q265P, H265Y *	[68,76,118]
	H266Y, H266L	[67,74,76]
	L267W, L267F	[73,86,88]
	T269P	[76,116]
	Q270P	[98]
	L271F, L271R	[86,113]
	G272D	[64]
	L274W	[88]
	Q277H, Q277R, Q277K	[84,86,88]
HATPase_c	N353Y	[115]
	P360Q	[84,87,95]
	H362N	[97]
	Y363F	[97]
	P377L	[84]
	F387Y	[115]
	S403F	[115]
	A408E	[72]
	R411del	[59]

TM1, first transmembrane domain (aa 10–29); PD, periplasmic domain (aa 29–142); TM2, second transmembrane domain (aa 142–164); HAMP domain, histidine kinases, adenylyl cyclases, methyl-binding proteins, and phosphatases (aa 145–214); HisKA, histidine kinase A domain (aa 218–280); HATPase_c, histidine kinase-like ATPase (aa 326–437) [51]. * indicates referent amino acids differed from amino acid at the same position in the PmrB protein of *A. baumannii* ATCC17978 (CP053098.1) [101,118].

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
