# Peer review of "Colistin Resistance in Acinetobacter baumannii: Molecular Mechanisms and Epidemiology"

_antibiotics, 2023, doi:10.3390/antibiotics12030516_

Round 1

Reviewer 1 Report

Summary: The article reviewed the molecular mechanisms of colistin-resistance in Acinetobacter baumannii, implicated mutations, and its global dissemination.

Title: with regard to the content of the review, the title should be considered to read "Epidemiology of colistin-resistant strains of Acinetobacter baumannii"

The body: Considering another recently published review in the area, the paper addressed specific point mutations within the molecular mechanisms of colistin resistance. Hence it addressed this specific gap.  

Conclusion: The author should include relevant conclusion section to the paper

Author Response

Response to Reviewer 1 Comments

Point 1: Title: with regard to the content of the review, the title should be considered to read "Epidemiology of colistin-resistant strains of Acinetobacter baumannii".

Response 1: Thank you for your suggestion. We have changed the title to “Colistin resistance in Acinetobacter baumannii: molecular mechanisms and epidemiology”.

Point 2: The author should include relevant conclusion section to the paper.

Response 2: Thank you for your suggestion. The conclusion section is included now.

Reviewer 2 Report

Colistin being a reserved antibiotic its use and studies on resistance is a matter of concern.  The topic, therefore, is of general interest, no doubt. Colistin resistance in Acinetobacter baumannii and epidemiology of colistin-resistant strains

This review is informative and it will help to understand colistin resistance in Acinetobacter baumannii.

1.       There is a minor spelling mistake in 49th line. Instead of feeding there should be bleeding (needs a thorough check to avoid both typos, meaning and grammatical correction)

2.       There should be more details in figure 1 and every illustration need to be made self explaining.

3.       In the introduction part there requires more information about colistin's chemical properties and variations. Because this will provide the information for understanding the chemical interactions at the time of action.

4.       There should tell more pictorial representation of each resistance mechanism.  

5.    Authors must also compare various resistance mechanisms with other aerobic pathogens such as; Pseudomonas and Stenotrophomonas as well as facultative anaerobes, e. g. Enterobacteriales (E. coli and Klebsiella at least).

Author Response

Response to Reviewer 2 Comments

Point 1: There is a minor spelling mistake in 49th line. Instead of feeding there should be bleeding (needs a thorough check to avoid both typos, meaning and grammatical correction).

Response 1: The observed mistake has been changed.

Point 2: There should be more details in figure 1 and every illustration need to be made self explaining.

Response 2: Thank you for the suggestion. We included more details in Figure 1 caption in order to make it self-explanatory.

Point 3: In the introduction part there requires more information about colistin's chemical properties and variations. Because this will provide the information for understanding the chemical interactions at the time of action.

Response 3: Thank you for the suggestion, we included requested data. Included data are about amphihilic surfaces of colistin and their role in interaction with bacterial membrane. This will make further mechanisms of action and mechanisms of resistance to colistin more understandable.

Point 4: There should tell more pictorial representation of each resistance mechanism. 

      Response 4: Thank you for the suggestion. We improved the representation of each resistance mechanism.

      Point 5: Authors must also compare various resistance mechanisms with other aerobic pathogens such as; Pseudomonas and Stenotrophomonas as well as facultative anaerobes, e. g. Enterobacteriales (E. coli and Klebsiella at least).

Response 5: Thank you for your suggestion. In section 3. we compared molecular mechanisms responsible for colistin resistance in different Gram-negative pathogens. We indicated similarities and differences compared to A. baumannii.

Reviewer 3 Report

This review aims to systematically present up-to-date knowledge on molecular mechanisms of colistin resistance in A. baumannii with a detailed review of implicated mutations and global dissemination of colistin-resistant strains.

Despite the public health importance of this review, there are areas of concern to be addressed by the authors.

Areas of concern:

Abstract:

Lines 14-15: This statement entails that this review is a systematic review, which is not the case. So, remove ‘’systematically’’.

Methodology

However, this is not a systematic review but the authors should indicate in the abstract whether the reviewed only published literature or both published and unpublished and how many papers were reviewed.

Introduction

The introduction does not provide explicit and comprehensive information regarding the justification for the conduct of the review in the context of what is already known. For example, the authors failed to justify why the focus of this review is on Colistin resistance in A baumannii knowing that other Gram-negative bacteria exhibit resistance to Colistin. The magnitude of this phenomenon should have been highlighted in this section. Moreover, other components of the review (sections 2 and 3) would have been announced in the introduction.

Section 3

Figure 2: This map should be colored and only shared ST should be inside the map. Other STs should be outside the map but their presence in the continents should be indicated by an arrow.

Moreover, the authors should indicate in this section the presence of A. baumannii STs in Australia/Oceania and should attempt to explain the absence of data from this continent.

Conclusion and recommendation

A section for conclusion and recommendations is missing in this review. This review is supposed to end up with some implications for practice or for research. More primary research should be recommended in the continents where no data were available.

Author Response

Response to Reviewer 3 Comments

Point 1: Lines 14-15: This statement entails that this review is a systematic review, which is not the case. So, remove ‘’systematically’’.

Response 1: Thank you for the suggestion. We introduced the changes according to your instructions.

Point 2: However, this is not a systematic review but the authors should indicate in the abstract whether the reviewed only published literature or both published and unpublished and how many papers were reviewed.

Response 2: Thank you for the suggestion. We introduced the changes according to your instructions.

Point 3: The introduction does not provide explicit and comprehensive information regarding the justification for the conduct of the review in the context of what is already known. For example, the authors failed to justify why the focus of this review is on Colistin resistance in A baumannii knowing that other Gram-negative bacteria exhibit resistance to Colistin. The magnitude of this phenomenon should have been highlighted in this section. Moreover, other components of the review (sections 2 and 3) would have been announced in the introduction.

Response 3: Thank you for your comment. We introduced the requested changes. Information regarding components of the review is now highlighted at the end of section 2, while information about the significance of colistin resistance in A. baumannii has been included at the end of section 2. These data clearly indicate and justify the focus on A. baumannii.

Point 4: Figure 2: This map should be colored and only shared ST should be inside the map. Other STs should be outside the map but their presence in the continents should be indicated by an arrow. 

      Response 4: Thank you for your comment. We used the freely available map of the world in SVG format in order to generate the picture, to our best knowledge we could not find one with adequate coloring that would match the data presented in this picture. We introduced the changes in order to highlight the absence of data for Australia and Oceania. Also, we included more details in the figure caption in order to eliminate eventual misguiding of the picture concerning the continents and presented data. We hope that these changes will meet your criteria.

      Point 5: Moreover, the authors should indicate in this section the presence of A. baumannii STs in Australia/Oceania and should attempt to explain the absence of data from this continent.

     Response 5: Thank you for the suggestion. We included a novel paragraph at the end of section 4 that gives an explanation for the lack of data from Australia and Oceania in this review.

     Point 6: A section for conclusion and recommendations is missing in this review. This review is supposed to end up with some implications for practice or for research. More primary research should be recommended in the continents where no data were available.

     Response 6: Thank you for the suggestion and useful guides for the conclusion section. We included this section in a revised version of the manuscript.

Reviewer 4 Report

It is a review about A. baumannii colistin resistance mechanisms and epidemiology of resistant strains. The subject is quite interesting since Acinetobacter infections represent a hot theme among clinicians, especially as regards hospital infections.

The paper is well conducted, however English need to be revised since there are some repetitions and often it is not so fluent. Furthermore, some paragraphs could result too long to read and they soul be better summarize.

In addition, I have the following concerns:

·         Line 29: Last resort antibiotic

·         Line 42: Thus, colistin is considered bactericidal

·         Line 57: Add a comma after “diabetes”

·         Line 59: Hospital length of stay

·         Line 60: What do you mean with “demographics”?

·         Line 63-65: Please rephrase since it is unclear

·         Figure 1: Please add letters or number in the single figures (1), 2) …) and fix the caption.

·         Please also talk about Acinetobacter as environmental bacteria and briefly report the difficulties in recognizing colonization from infections

·         I suggest briefly mentioning solutions to colistin resistance such as new drugs (cefiderocol, durlobactam/sulbactam) and old drugs (such as fosfomycin combinations). These are some references you could find helpful for the manuscript 10.1128/spectrum.02347-22, 10.3390/antibiotics12010049, 10.1128/aac.00781-22

·         Paragraph 3.2.1 is too long, be more concise

Author Response

Response to Reviewer 4 Comments

     Point 1: The paper is well conducted, however English need to be revised since there are some repetitions and often it is not so fluent. Furthermore, some paragraphs could result too long to read and they soul be better summarize.

Response 1: Thank you for the suggestion. The usage of the English language has been revised and improved.

Point 2: Line 29: Last resort antibiotic; Line 42: Thus, colistin is considered bactericidal; Line 59: Hospital length of stay.

Response 2: The indicated changes have been introduced.

Point 3: Line 57: Add a comma after “diabetes”.

Response 3: In this case, this is related to diabetic kidney disease, a type of kidney disease caused by diabetes. Diabetic nephropathy is a serious complication of type 1 and type 2 diabetes, and for example, in the USA about 1 of 3 people living with diabetes have diabetic nephropathy.

Point 4: Line 60: What do you mean with “demographics”?

Response 4: Patient demographics are a patient’s basic information, including age, biological sex, gender, ethnicity etc.

Point 5: Line 63-65: Please rephrase since it is unclear.

Response 5: This sentence has been rephrased to be more clear now.

Point 6: Figure 1: Please add letters or number in the single figures (1), 2) …) and fix the caption.

Response 6: Thank you for the suggestion. This Figure has been changed according to your instructions.

Point 7: Please also talk about Acinetobacter as environmental bacteria and briefly report the difficulties in recognizing colonization from infections

Response 7: Thank you for the suggestion. We included a paragraph within section 2 that addresses Acinetobacter as environmental bacteria, as well as relevant references.

Point 8: I suggest briefly mentioning solutions to colistin resistance such as new drugs (cefiderocol, durlobactam/sulbactam) and old drugs (such as fosfomycin combinations). These are some references you could find helpful for the manuscript 10.1128/spectrum.02347-22, 10.3390/antibiotics12010049, 10.1128/aac.00781-22

Response 8: Thank you for the suggestion. We included that information within the conclusion section as promising novel therapies.

Point 9: Paragraph 3.2.1 is too long, be more concise

Response 9: Thank you for your suggestion. We introduced the changes in order to present this comprehensive issue in a more concise manner.

Round 2

Reviewer 2 Report

Changes incorporated are sufficient on the scientific ground.